# Functional *Cyperus esculentus* L. Cookies Enriched with the Probiotic Strain *Lacticaseibacillus rhamnosus* SL42

**DOI:** 10.3390/foods13162541

**Published:** 2024-08-15

**Authors:** Noussaiba Belmadani, Wafa Kassous, Kawtar Keddar, Lamia Amtout, Djahira Hamed, Zohra Douma-Bouthiba, Vlad Costache, Philippe Gérard, Hasnia Ziar

**Affiliations:** 1ProbiotSanté Team, Laboratoire des Micro-Organismes Bénéfiques, des Aliments Fonctionnels et de la Santé (LMBAFS), Abdelhamid Ibn Badis University, Hocine Hamadou Street, Mostaganem 27000, Algeria; noussaiba.belmadani.etu@univ-mosta.dz (N.B.); kawtar.keddar@univ-mosta.dz (K.K.); amtoutlamia@gmail.com (L.A.); djahira2004@yahoo.fr (D.H.); doumabouthibazohra@gmail.com (Z.D.-B.); 2MIMA2 Imaging Core Facility, Micalis Institute, INRAE, 78350 Jouy-en-Josas, France; vlad.costache@inrae.fr; 3Micalis Institute, INRAE, AgroParisTech, Paris-Saclay University, 78350 Jouy-en-Josas, France; philippe.gerard@inrae.fr

**Keywords:** functional cookie, organic tiger nut flour, probiotic strain SL42, physical analysis, sensorial analysis

## Abstract

This study presents for the first time functional cookies for diabetics made with 100% organic *Cyperus esculentus* L. flour, either plain or enhanced with 5% polyfloral honey syrup containing the probiotic strain *Lacticaseibacillus rhamnosus* SL42. The flour’s chemical composition and rheological and functional properties were analyzed, and 33 diabetic and semi-naive panelists assessed the cookies’ sensory properties. MRS-cys agar cultures and SEM analysis evaluated SL42 survival and adhesion capacity over 21 days at 25 °C. Results showed that the flour and its extracts are rich in polyphenols and flavonoids, indicating strong antioxidant and antibacterial properties. Both cookie types met international standards when compared to commercial cookies and had similar physical properties. Sensory evaluation on day 1 revealed higher quality indicators for honey syrup-enriched cookies, but after 15 days, control cookies were preferred. The CIE LAB analysis confirmed the dietetic flour’s typical dark color, with honey syrup-enriched cookies being darker. Despite textural differences, both cookies maintained detectable crispness over storage. Honey syrup-enriched cookies effectively carried *L. rhamnosus* SL42, remaining viable at 6.43 Log CFU per cookie after 21 days and adhering to the cookie’s surface, as confirmed by SEM analysis. Further research is recommended to better understand the therapeutic value of these cookies.

## 1. Introduction

The human need for food is not just a choice but rather an essential part of the body. In recent years, there has been a growing emphasis on promoting health by lowering the glycemic index (GI) of meals, particularly those high in sugars and starch, like biscuits. To reduce GI, one popular solution is to substitute sucrose with natural sweeteners. However, substituting, partially or totally, wheat flour content with a dietetic one in the product composition is also an effective option.

*Cyperus esculentus* L. is a grass-like plant from the Cyperaceae family. Its little, delicious tubers mature underground. *Cyperus esculentus* L., also known as tiger nuts, terrestrial almonds in Africa, chufa in Spain, and Hab Alaziz in Algeria and in many Arab countries, is considered one of the ancient plants that was famous for its use in many fields, such as food and traditional medicine, thanks to the nutrients and important components that are present in it [1].

The use of tiger nuts dates back more than 3000 years and is already mentioned in the writings of ancient Egypt. It was introduced to Europe by the Arabs in the 19th century, and it is especially in kitchens that the use of tiger nuts is traditionally reserved. It can be eaten cooked or grilled. In several countries, it is reduced to powder and is incorporated into the composition of porridge or cookies. In Spain, France, the Dominican Republic, Mexico, Panama, and the USA, Horchata de chufa is a nutritious milky product obtained mechanically by aqueous extraction using pressure from tiger nuts [2].

Tiger nut products (flour, oil, and milk with and without added sucrose) have significant amounts of bioactive substances that are beneficial to macular health. Tiger nut milks revealed higher protein digestibility than flour, whereas the latter would be appropriate for dietary recommendations for hyperglycemic patients. As a result, flour could be an excellent element for diabetic-friendly functional food compositions [3].

In several studies, tiger nut, proposed for the management of type 2 diabetes [4], showed anti-inflammatory and anti-apoptotic effects to prevent testicular dysfunction [5], improved male arousal [6], reduced diarrheal symptoms in albino rats [7], and reduced oxidative stress in the liver and inflammation with atherosclerosis [8]. The results are most likely due to the presence of alkaloids, quercetin, vitamins, steroids, zinc, and other compounds in tiger nuts. As a result, in addition to being food and industrial material, tiger nuts offer the potential to be transformed into functional foods.

Biscuits, or cookies, are chemically leavened baked goods made with wheat flour, sugar, shortening, and a little water [9]. Cookies are the most popular cereal products consumed around the world due to various reasons, including their taste quality, affordable cost, availability in different varieties, and their long shelf-life. Dietary cookies, such as those without sugar or salt, with protein, high in insoluble fibers, or gluten-free, are labeled “healthy”, “light”, or “low-fat”. This range is constantly expanding and multiplying nutritional promises [10].

Cereal-based baked foods (CBBFs) are often consumed and can contain useful ingredients such as probiotics. Probiotic cereal-based products are an alternative to traditional probiotic dairy foods, and they do not require refrigeration during distribution and storage. The fundamental problem in designing a probiotic cereal-based product is keeping the bacteria alive. Thus, probiotics could be incorporated into CBBF by several methods, including syrups, microencapsulated edible films, chocolate toppings, and spore-forming microorganisms [11].

DNA damage has been linked to type 2 diabetes and its complications, particularly oxidative stress [12]. Tiger nut sugar may have a cytoprotective effect, mainly on hepatocytes. High content of vitamins C and E, as well as phenolic acids, may explain its considerably larger cytoprotective impact. In vitro, such elements have been shown to have notable antioxidant activity, and to protect against induced DNA oxidation by decreasing DNA scission [13]. Furthermore, tiger nut powder, which has a comparable lipid profile to olive oil [3], is an intriguing material for the production of healthful products for diabetics or people with high blood pressure based on Algerian folk medicine.

On the other hand, probiotic supplementation has been shown to reduce oxidative stress and increase antioxidant indicators in numerous interventional investigations [12]. In the present study, dietary cookies were prepared using an organic tiger nut flour instead of wheat flour. The aim was to use tiger nut flour to make healthy and sugar-free cookies for diabetics. We present here for the first time cookies with or without honeybee syrup containing a spry-dried probiotic bacterium SL42. *Lacticaseibacillus rhamnosus* strain SL42, isolated from breast milk, is an interesting isolate from our laboratory and was able to abolish bacterial translocation by reinforcement of the gut barrier and manage the anti-inflammatory responses [14]. The intention of this research is to make functional cookies for diabetics from tiger nut flour. The outcomes of this study may contribute to a better understanding of the design and manufacture of healthy sugar-free cookies, enhanced or not with honeybee syrup as a probiotic carrier, as well as valuable guidance for improving the added value of tiger nuts and future applications of CBBF in the food industry.

Thus, we first determined the physico-chemical, rheological, and functional properties of the organic flour that will be used for making the cookies. Hence, under storage conditions, sensory analyses of plain cookies or cookies enriched with honey syrup were also carried out using a diabetic panel. Finally, the probiotic survival and adherence were assessed.

## 2. Materials and Methods

### 2.1. Raw Materials

The plant material consists of tubers of organic tiger nut (Figure 1), procured from herbalists in Southern Algeria, but originating from the Eastern Tchad, more specifically the Ouaddaï region.

### 2.2. Reducing Tiger Nuts to Flour

The tubers with the greatest homogeneity in size, maturity, and weight were selected for making tiger nut flour. The process of obtaining the flour is described in Figure 2. Following drying, the dried tubers were passed through a spray mill to obtain particles of 300 μm. The flour obtained was packaged in polyethylene plastic bags.

### 2.3. Chemical Description of Raw Tiger Nut Flour

Moisture (air oven method), protein (crude protein combustion method), fat (crude fat method), carbohydrates (proximal differences), fiber (total dietary fiber method), mineral (atomic absorption spectrophotometry method), and ash (incineration basic method) contents (%) of the flour were determined according to the American Association of Cereal Chemists (AACC) International official methods 44–15.02, 46–30.01, 30–10.01, 32–45–01, 40–70.01, and 08–03.01 [15], respectively.

Glucose, fructose, and sucrose contents were analyzed by ion chromatography (IC). All chemicals and reagents used were at least of analytical reagent grade. D (−)-fructose, D (+)-glucose anhydrous, and D (+)-sucrose monohydrate were purchased from Fluka (Switzerland). All aqueous solutions were prepared using deionized water from RiOs™ type I simplicity 185 (Millipore Waters, Milford, MA, USA) with a resistivity of 18.2 MΩ cm.

Chromatographic separations were performed on a Dionex Instrument DX-500 IC system (Sunnyvale, CA, USA). The system consisted of a GP40 gradient pump and ED40 electrochemical detector equipped with a thin-layer-type amperometric cell. The cell consisted of a 1 mm diameter gold working electrode and platinum counter electrode in IPAD mode. The separations were carried out on a CarboPac PA 10 column set consisting of a guard column (50 mm × 2 mm ID) and an analytical column (250 mm × 2 mm ID). The sample injection volume was 25 μL. The flow rate was 1 mL min^−1^. The columns were placed inside a temperature controller. Chromatographic system control, data acquisition, and analysis were performed using PeakNet 6.0 software (Dionex).

An accurate amount of raw flour (0.25 g) was dissolved in 25 mL of water. Sample solutions were filtered through a 0.22 μm membrane filter. The sample was diluted 100-fold with water and was directly injected into the chromatographic system for the analysis of sugars [16].

### 2.4. Rheological Characteristics of Tiger Nut Flour

#### 2.4.1. Bulk Density (BD)

The bulk density was measured using the Wang and Kinsella [17] method. A total of 10 g of the flour was placed in a 25 mL graduated cylinder and smoothly tapped ten times on a bench top from a height of 5–8 cm. The final volume of the test flour was measured, and BD was represented in grams per milliliter.

#### 2.4.2. Water Absorption Index (WAI)

WAI was calculated using the method previously described for cereals [18]. The powdered flour was suspended in water at room temperature for 30 min, gently mixed during that time, and then centrifuged at 3000× *g* for 15 min. The supernatants were decanted into a pre-weighed evaporating dish. WAI was the weight of gel obtained after removing the supernatant per unit weight of the original dry solids.

#### 2.4.3. Oil Absorption Capacity (OAC)

The oil absorption capacity was calculated using the method of Lin et al. [19]. In a pre-weighed centrifugal tube, 0.5 g flour and 10 mL refined oil were combined and vortexed for 10 min. The tubes were centrifuged for 25 min at 3000× *g*. After inverting for 10 min, the oil was drained and the centrifuge tubes were weighed.

#### 2.4.4. Foaming Capacity

The Lin et al. [19] technique was used to measure the foaming capacity. Five percent (5%) of 100 mL of flour dispersed in water was homogenized. The foam volume was measured immediately after the mixture had been transferred to a 2500 mL graduated measuring cylinder. Foaming activity was indicated as a percentage increase in volume.

#### 2.4.5. Gluten Yield

The hand-washing method was performed according to AACC International Method 38-10.01 [15]. Aliquots of 25 g of flour were mixed with an appropriate amount of water (12–15 mL), kneaded until the resulting dough was firm and smooth, and then rested for 2 h to allow the gluten structure to develop. The dough was washed until just the dark gluten ball remained, after which the wet gluten was weighed.
Wet gluten yield = (weight of wet gluten obtained/weight of flour) × 100(1)

#### 2.4.6. X-ray Diffraction Analysis (XRD)

A total of 2 mg of flour was measured by automatic multifunctional X-ray diffractometer (D8 Advance, Bruker, Germany) [20]. The X-ray generator was run at 40 kV and 40 mA, a scanning range of 4–60° 2θ, and a scan speed of 2.0°/min.

### 2.5. Functional Properties of the Flour

Aqueous (using 100% ultrapure water) and hydro-ethanolic (30:70; water:ethanol, *v*/*v*) tiger nut flour extracts were subjected to the following analysis.

#### 2.5.1. Polyphenol Content

Each extract (500 μL) was mixed with 2.5 mL of 10% Folin–Ciocalteu reagent and 2 mL of 7.5% saturated Na_2_CO_3_ [21]. The reaction was conducted in a water bath at 50 °C for 15 min, and the absorbance was measured at 760 nm using a UV spectrophotometer (UV-1800, Shimadzu Corporation, Kyoto, Japan). Results were presented as Gallic acid equivalents (GAE, mg/100 g extract) (y = 0.0287x − 0.1374; R^2^ = 0.9712).

#### 2.5.2. Flavonoid Content

A total of 1 mL of the extract was mixed with 0.3 mL of 5% NaNO_2_ and 10% AlCl_3_. After 6 min, 2 mL of 1 M NaOH and 2.5 mL of distilled water were added. The color intensity of flavonoids [21] was measured at 510 nm (UV-VIS spectrophotometer, Shimadzu Corporation, Kyoto, Japan). The total flavonoid content was calculated using quercetin equivalent (QE) and expressed as mg/g (y = 0.0431x – 0.0682; R^2^ = 0.9987).

#### 2.5.3. Antioxidant Activity

The Brand-Williams et al. [22] method was employed, using a spectrophotometer (DU^®^ 730, Beckman Coulter Inc., Brea, CA, USA). Trolox was used to generate a calibration line ranging from 0 to 200 mg/L for the assessment of antioxidant activities in both methods. The results are presented as IC_50_ and EC_50_ values (mg/mL).

#### 2.5.4. Antimicrobial Activity

For the antimicrobial activity, 100 μL of the microbial suspensions of *Candida albicans* ATCC 10231, *Staphylococcus aureus* ATCC 33862, *Bacillus subtilis* ATCC 6051, *Bacillus cereus* ATCC 10876, *Escherichia coli* ATCC 25922, and *Pseudomonas aeruginosa* ATCC 27853 was cultivated on plates containing Mueller–Hinton agar (Difco) in triplicate. Sterile filter paper disks impregnated with the tiger nut flour extracts (100 mg/mL: hydroethanolic or aqueous extracts) were placed on the surface of the inoculated medium using a sterile forceps. A sterilized disc was used as a negative control, and amoxicillin (80 mg/mL) was used as a positive control. The results were expressed as the diameters of the inhibition zone in millimeters, formed around the discs containing the extracts after incubation of the plates for 24 h at 37 °C.

### 2.6. Preparing Healthy Cookies

#### 2.6.1. Formulation

A 100% organic tiger nut flour was used to make cookies. Cookies were prepared in our laboratory. In the formulated cookie dough, sucrose was not added.

The cookies were prepared with a slight modification of the AACC standard method (10e50D) [23] using the following ingredients: organic tiger nut flour (100 g), sodium bicarbonate (1 g), salt (1 g), skimmed milk powder (20 g), butter (50 g), and water (20 mL). Firstly, butter and skimmed milk powder were mixed to form a cream, and then the mixture of flour, sodium bicarbonate, and salt was added and carefully mixed to form a dough. Then, the dough was crushed and spread to a uniform thickness of 0.5 cm and cut into circular shapes 5 cm in diameter. After cooking at 170 °C for 15 min, cookie samples were cooled. Cookies without (control plain cookies, CC) or containing 5% honeybee syrup (CH) were subjected to the following analyses. The CH cookies were sugar-free cookies impregnated with a 5% (*w*/*v*) cooled and pasteurized (80 °C/30 min) honeybee (Eucalyptus and Greenbrier polyfloral honey from a local beekeeper, Chlef city, Algeria) syrup (amounts in 100 g: water 16.5 g, total carbohydrates 66.2 g, glucose 26 g, fructose 36 g, sucrose 0.25 g) containing the *Lacticaseibacillus rhamnosus* SL42 (GenBank: OQ300076.1) probiotic strain (LMBAFS laboratory’s collection, Mostaganem University, Algeria) to make dietary cookies containing probiotics.

The spray-dried powder of *L. rhamnosus* SL42 was added to a final concentration of 1 × 10^9^ CFU/mL in cooled syrup and 1 mL was swirled on each cookie.

#### 2.6.2. Spray-Drying of SL42

The spray-dried powder of *L. rhamnosus* SL42 was prepared following the method described by Sompach et al. [24] with slight modification. Sucrose was employed as the encapsulant formulation solution for spray-dried microcapsules, with a concentration of 5% (*w/w*) in deionized water. The sucrose solution was agitated for 15 min at 600 rpm with an overhead stirrer (RW 20 digital, IKA, Taufkirchen, Germany). The concentrated cells were disseminated into the solution and agitated for a further 30 min. The feed suspension was spray-dried using a mini spray dryer (BÜCHI Labortechnik AG, Flawil, Switzerland) and continuously stirred with a magnetic stirrer to maintain homogeneity; the spray drying was carried out at a constant feed flow rate of 7 mL/min and an inlet air temperature of 130 °C. The aspiration rate was kept at around 35 m^3^/h, while the flow meter spraying flow rate was 475 L/h [24]. The spray-dried microcapsules were removed from the collection vessel and placed in a zip-lock polyethylene bag, which was then packed with aluminum foil laminate bags made of three layers of polyethylene terephthalate, aluminum foil, and polyethylene.

#### 2.6.3. Storage of Cookies and Assessment of Flour and Honey Effects on SL42

Both cookies were packed individually into light impermeable packaging and stored at 25 °C in a controlled temperature chamber under 75% relative humidity.

Before cookie preparation, honeybee (5%, *w*/*v*) or organic tiger nut flour (0 to 2%, *w*/*v*) solutions were individually added to SL42 culture, and checked for any bacterial inhibitory effect.

### 2.7. Appearance and Texture Analysis

The diameter and thickness were measured using a digital caliper at two different locations of each cookie, and the average was calculated for each of them. The fracturability was measured by exercising the same force at the cookie’s center. The average of six cookies was recorded for each batch, while the weight of the cookies was determined using an electronic scale. The spreading ratio was calculated by dividing the cookie diameter by its height. More than six cookies were used to test hardness (highest peak force) in each sample using a texture analyzer (TA-XT2i, Stable Micro Systems, UK). Pre-test, test, and post-test speeds were 1.5, 2, and 10 mm/s, respectively. The peak force required to break the cookies was defined as fracture force (N). Commercial wheat flour cookies (Delight^®^ from Algerian market, Douera, Algiers) were used as reference. 

Cookies’ color characteristics (L*, a*, and b* of the CIE system) were measured with a chromameter (CR-400 Series, Konica Minolta, Inc., Tokyo, Japan). Three samples were used, and six measurements were taken and averaged for each specimen.

### 2.8. Scanning Electronic Microscopy Analysis of Cookies

Morphological properties of both cookies, the prepared syrup, and SL42 adhesion were evaluated by scanning electron microscopy (FEG-SEM-LV SU-5000, Hitachi, Milexia, 91190-Saint-Aubin, France). A part of each cookie was directly observed using the environmental configuration (50 Pa, UVD detector, 10 keV, 30 spot size, Peltier stage system at −15 °C) and another part of the same cookie was fixed for high-resolution observations (SE(L) detector, 2 keV, 30 spot size, 5 mm WD). Various magnifications were used from 100 to 25,000 X. For fixing, samples were immersed in a solution of 2% glutaraldehyde and 0.1 M sodium cacodylate buffer, pH 7.4. The culture within the syrup was deposited on a 1 cm square microscopy glass slide, cut with a diamond pen. Glass slides were previously cleaned with 70% ethanol, 5 min plasma cleaner, 10 min sonication bath in ethanol 100% and in MilliQ water, and finally coated with 0.1 M polylysine (Merck Sigma Aldrich) over the course of 5 min, then rinsed with MilliQ water. Samples were incubated in the fixative solution for one hour at room temperature, and then kept overnight at 4 °C. The fixative solution was removed, and samples were rinsed two times for 10 min in 0.2 M sodium cacodylate solution (pH 7.4). The samples underwent progressive dehydration by soaking in a series of ethanol baths before critical-point drying under CO_2_ (Leica EM300, slow 20 exchange cycles, 2 min delay). Samples were mounted on aluminum stubs (15 mm and 32 mm diameter) with carbon adhesive discs (LFG France) and were coated with Au/Pd (Quorum SC7620, 5 Pa of Ar, 3 × 180 s of sputtering at 3.5 mA).

### 2.9. Sensorial Evaluation

All of the sensorial sessions (UAIBM-DSA-LMBAFS-N°081-IRB Approval) took place before lunch (11:00–13:00). The study included 33 diabetic and semi-naïve panelists (aged 40–65) who commonly consumed cookies. The panelists examined all six samples from each cookie type (CC or CH) in a single session. Cookies were coded with random three-digit numbers using a balanced complete block experimental design. For the general evaluation, specific sensorial attributes of taste, color, texture, aroma, and overall acceptability were evaluated. For the hedonic test, taste/flavor liking, smell liking, color liking, and texture liking were evaluated using a nine-point scale (1 = extremely dislike to 9 = extremely like).

### 2.10. Statistical Analysis

Data were analyzed using IBM (SPSS statistics 26, Chicago, IL, USA). All experiments were independently performed in three replicates and the results are reported as the mean values ± standard deviation. Duncan’s test was used to test for group differences. *p*-values < 0.05 were considered significant.

## 3. Results

### 3.1. Physico-Chemical Parameters Demonstrated a Rich and Nutritious Flour

The approximate centesimal composition of the tiger nut flour (Table 1) showed water and ash contents of 8.3% and 3.04%, and high mineral and fat contents of 2.26% and 26.54%, respectively. The latter value falls within the specific range of this plant. The contents of protein, carbohydrate, and fiber on a dry basis were also high.

Data from IC chromatography revealed that tiger nut flour had a considerably higher concentration of sucrose (~15 g/100 g dry basis) than glucose and fructose (Figure 3), confirming its natural and desired sweetness.

### 3.2. Organic Tiger Nut Flour Had Good Rheological Values

The tiger nut flour exhibited a bulk density of 0.74 g/mL and a dispersibility of around 68% (Table 2). Water absorption capacity was low (0.08 g/mL), while oil absorption (2.2 mL/g) and foaming capacities (12%) were high. Thus, the result in Table 2 confirmed that tiger nut flour is gluten-free.

The XRD pattern of the whole tiger nut flour is shown in Figure 4. As can be seen from this figure, the X-ray diffraction pattern at 2θ is 8.1°, 16.9°, 18.6°, and 22.4°, indicating that the tiger nut starch in the flour is a C-type polymorphic structure. The three crystalline peaks that occurred in the range 10° to 25° 2θ represented the B-type tiger nut starch.

The tiger nut flour’s structure has also a weak single diffraction peak at 19.2°, and two sharp diffraction peaks at 34.5° and 35.3°, suggesting the existence of long arrangement structure.

### 3.3. Prosperous Functional Qualities of Tiger Nuts

The total polyphenol content (Appendix A) in the aqueous extract was higher (*p* < 0.05) than that of the ethanolic extract (water 30v/ethanol 70v), with registered values of 111.82 ± 0.2 and 68.5 ± 0.2 mg EGA/100 g DW, respectively. Water appears to be more effective than organic solvents in extracting phenolic compounds from tubers, implying high content of water-soluble phenolic chemicals.

The aqueous extract (280 ± 0.18 mg EQER/100 g DW) contains slightly (*p* > 0.05) more flavonoids than the ethanolic extract (198.18 ± 0.3 mg EQER/100 g DW) (Appendix A).

As can be seen from Figure 5, the aqueous and ethanolic extracts of *C. esculentus* have antioxidant activities with respective IC_50_ values of 49.96 ± 0.08 and 6.22 ± 0.07 mg/mL, as expressed by the DPPH method. The one from the ethanolic extract was obviously higher (*p* < 0.05).

The results of the FRAP (Ferric Reducing Antioxidant Power) method (Figure 6) showed that the aqueous and ethanolic extracts of *C. esculentus* have antioxidant activities with respective EC_50_ values of 211.5 ± 0.2 and 12.15 ± 0.05 mg/mL, as expressed by the FRAP method. The ethanolic extract, compared to the aqueous extract, provides a significant (*p* < 0.05) inhibition on iron ions.

The results on the antimicrobial properties of aqueous and ethanolic extracts of 100 mg/mL of tiger nuts revealed a wide spectrum of action against potentially pathogenic microorganisms. The inhibition zones observed around the discs impregnated with the extracts were measured using a caliper. The results are presented in Appendix A. Both aqueous and ethanolic extracts showed inhibitory actions against the tested pathogens. Indeed, the 30v water/70v ethanol extract exerted an inhibitory activity two folds higher on average than that conferred by the water-soluble constituents, except against *Bacillus* species.

### 3.4. Both Cookies Follow the International Standards

We were able to bake sugar-free dietary cookies using 100% organic tiger nut flour. The photographs in Figure 7 and Appendix A show the overall appearance of the baked cookies as well as the macroscopic changes between control plain cookies and those with honeybee syrup containing the SL42 probiotic strain.

Based on the results obtained in Table 3, it is noted that there is a significant (*p* < 0.05) difference in weight, diameter, thickness, and spread ratio between the plain tiger nut cookie and the one with honey and probiotic SL42 strain.

The comparative analysis of the tiger nut cookies and the commercial one based on wheat flour and cocoa (Delight©) revealed the “fulfilment” of the technological criteria in weight, diameter, and thickness in baked cookies. Moreover, tiger nut cookies were less (*p* < 0.05) vulnerable to breaks (Appendix A).

### 3.5. The Probiotic Strain SL42 Was Able to Grow in the Presence of Tiger Nut Flour or Honeybee Syrup

Cultural trials on the SL42 probiotic strain revealed its good growth capacities (Appendix A) in the presence of tiger nut flour (up to 2%) and honeybee syrup (5%, *w*/*v*).

### 3.6. Dark Color of Tiger Nut Cookies Revealed by CIE System Analysis

The color of cookies affects people’s first acceptance of these food items. Table 4 and Table 5 show the trichromatic analysis results for control plain cookies vs. those with honey and probiotics after 1 and 15 days of storage at 25 °C, respectively.

The L* or lightness of the stored cookies was low, ranging from 61.75 to 62.75, reflecting their dark color (*p* < 0.05). In the current study, the baking of raw (non-peeled) tiger nut flour cookies resulted in dark cookies with a chromatic characteristic a* of around 7. In contrast, the b* chromatic characteristic appears to be higher (*p* < 0.05) in plain cookies (25.25) than in those with honey and probiotics (23.25).

### 3.7. Sensory Evaluation of Both Tiger Nut Cookies Was Positive

Table 6 and Table 7 show the overall sensory analysis values of control cookies and probiotic-enriched honey syrup cookies baked from tiger nut flour after 1 day and 15 days of storage at 25 °C, respectively. The first storage day findings revealed that cookies enhanced with honey syrup (CH) scored higher (*p* < 0.05) on sensory quality parameters such as aroma, color, and taste, as well as overall acceptability, than control cookies (CC). After 15 days of storage, the appreciation had reversed in favor of control cookies (*p* < 0.05).

On the first day, the difference focused on the overly dark hue of the honey syrup-enriched cookies, as well as their overpowering sweetness, according to several panelists. Following 15 days of storage, the aroma criterion was added to the differences (*p* < 0.05) exposed between the two types of cookies. On that time-point, our panelists rated control cookies higher (*p* < 0.05).

### 3.8. Panel Opinion, Level of Satisfaction, and Hedonic Test

The first characteristic evaluated was taste and flavor. After one day of storage, 50% of panelists rated the flavor as excellent (*p* < 0.05) for control cookies, compared to 65% for honey-rich cookies. In fact, all of our panelists described both cookie types as having the product’s typical taste, with no negative connotations recorded (Figure 8).

The appreciation for control cookies remained unchanged (55%) at the second tasting session (*p* > 0.05), while those supplemented with honey were more than harmonious (45%) or excellent (45%) (Figure 9). Its distinctive smell had the same outcome as the taste and flavor. Proportions of 75% and 85% of our panelists (*p* < 0.05) found that our control or honey-enriched cookies had a good or excellent smell, similar to freshly baked products.

After 15 days of storage, the panellists’ positive assessment remained consistent (85 to 95%) because our cookies retained or even increased their extremely nice smell. This could be mostly due to the type of the chosen packaging.

In terms of color perception, our tasters found the honey syrup-soaked cookies darker (D1/D15: 40%/25%) than the control cookies (D1/D15: 20%/10%), despite the fact that this difference was expected following storage. This could possibly be attributable to the type and composition of the polyfloral honey utilized. Furthermore, our cookies had a 60% color match to the desired product, with no negative connotations.

Honey-enriched cookies had a softer texture after one day of storage (*p*< 0.05). However, disparities in hardness values between the two cookie groups diminished afterward, resulting in equivalent observations (*p* > 0.05) on the 15th day (Figure 8 and Figure 9).

Our cookies were rated as “crispier” by 50–55% of our tasters and “well baked” by 30–35%. Only 10 to 15% of our tasters complained that our cookies were “too soft or hard” after 15 days of storage. Sensory examination of our fresh stored cookies (1 day of storage) found that the majority of tasters preferred the cookies made with tiger nut flour. Only 5% of our panel “dislike slightly” the baked cookies (see Figure 10). This overall displeasure subsided after 15 days of storage (Figure 10). In fact, 35 and 25% of our panelists liked “extremely” and “very much” the control cookie, and 25 and 40% the one with honey bee syrup, respectively.

The sensory analysis results should be linked to a sequence of biological events that occur during the cookie manufacturing and storage process. These responses cause changes in color, texture, aroma, and taste.

### 3.9. L. rhamnosus SL42 Survived at Recommended Probiotic Level in Stored Cookies

Table 8 illustrates the bacterial load of the probiotic strain *L. rhamnosus* SL42 on a single stored cookie. This bacterium has been introduced in honey syrup, which already provides the bacterium stability when stored at room temperature. After 21 days’ storage, the load of *L. rhamnosus* was 6.43 Log CFU/one cookie.

### 3.10. SEM Displays Distinguished Morphology between the Two Types of Cookies and Confirms SL42 Adhesion

Scanning Electron Microscopy (SEM) images are shown in Figure 11. Figure 11a showed the SEM micrographs of the spray-dried SL42 cells within sucrose biofilm in honey syrup.

As shown in Figure 11b, the starch granules were prominent, with a morphology that changed from smooth, round, and without fractures in control cookies (CC), to granules of starch with fracture and more severe adhesion in honeybee syrup-enriched cookies (CH).

The syrup clearly influenced the morphological properties of the tiger nut cookies. The subsequent cookies (Figure 11c and Appendix A) include a tiny amount of tiger nut starch in the shape of spherical granules, the majority of which are irregular flaky structures with threads on their surfaces. In general, syrup addition has left aggregates with enlarged particle size and holes.

On the other hand, *L. rhamnosus* SL42 was added as a spray-dried powder in the honey bee syrup, which adhered to the cookie surface (Figure 11c and Appendix A), reflecting a good capacity to withstand a temperature of 25 °C and a low humidity in stored cookies, as confirmed above by MRS-cys culture (Table 8).

## 4. Discussion

Since ancient times, humans have used the plants that grow around them to nourish and to treat themselves. It is time to rediscover these plants that have been forgotten for too long, and we can take advantage of their many virtues in our daily lives.

For centuries, our ancestors accumulated real knowledge about the medicinal qualities of plants. Indeed, the plant kingdom is an inexhaustible source of molecules with therapeutic interest. In this context, much of the interest of the current research focuses on the study of secondary metabolites, which are often active ingredients in medicinal plants, and the assessment of their therapeutic values on which the pharmaceutical industry relies extensively for the development of new medicines.

Tiger nuts or *Cyperus esculentus* L. are multi-faceted therapeutic tubers, as recognized in the traditional Algerian pharmacopoeia, especially its aqueous extract for treating hyperglycemia and high blood pressure.

In the present work, we first explored its physico-chemical composition and tested its bioactive components as antioxidants and its inhibitory effect on microorganisms’ responsible for toxic infections. Secondly, we proposed a formulation of cookies without the addition of sugar and with 100% whole organic tiger nut flour. The two types of cookies proposed fit into the range of dietary cookies.

Our results showed that the physico-chemical composition of tiger nut flour reflected a rich and nutritious meal. The water content was lower than that determined by Oldele and Aina (3.5–3.75%) [25] and comparable to that (8.03–8.6%) highlighted by Yapi et al. [26] when studying different varieties of tiger nut flours. The ash content was higher compared to the values reported in the literature: 2.17% by Bankoffi et al. [27] and 1.6–2.4% by Yapi et al. [26]. The high ash content (Table 1) indicates a high content of minerals (2.26%), such as sodium, calcium, iron, magnesium, phosphorus, and zinc, which are more abundant in the *C. esculentus* tubers than in commonly consumed cereals [27]. The ash content also indicates that the tuber is free of contaminants and is healthy to eat [28].

Tiger nut flour was also rich in fat and minerals. Oldele and Aina [25] found a fat content ranging from 32.13 to 35.43%, higher compared to that found in our study. In contrast, Nina et al. [29] found lower values ranging from 19% to 22%. However, the fat content is considered to be higher when comparing the tiger nut flour to common millet (7.6%), quinoa (6.3%), cajan peas (1.80%) [30], and wheat flour (3.10%) [31].

Tiger nuts are rich in nutrients along with a fat composition comparable to olive oil and a high concentration of minerals, particularly potassium and phosphorus. The crude fat content explains its significance in a balanced diet, since it lowers the risk of heart disease and lowers cholesterol levels, among other benefits [32].

In general, all nutrient contents were similar and consistent with those reported in the literature [1,33].

Ion chromatography is a useful analytical method with numerous applications in food analysis, allowing food scientists to quantify many major groups of analytes, such as sugars, that are critical to human health. The data acquired by IC are recognized as highly precise and accurate, and can be used for a full monitoring of the composition of food products [34].

The tiger nut flour contains higher sucrose concentration compared to its monosaccharides glucose and fructose. Unfortunately, the high sucrose content of tiger nut meal has received little attention in recent tiger nut investigations. The lack of studies on its sucrose content is a significant impediment to the full exploitation of tiger nut as a potentially valuable resource being discarded [13]. Given its considerable sucrose content, tiger nut meal could become a new non-centrifugal sugar resource, which would be one way to increase its value [13].

Furthermore, unrefined sugar, such as tiger nuts, may be considered a healthy food, since they contain several non-sucrose components, including fructose, phenols, flavonoids, minerals, and vitamins. Several studies have shown that unrefined sugars have positive benefits, such as immunological stimulation and anticancer, antioxidant, and cytoprotective effects [35,36].

In the present study, we also checked the rheological aspects of tiger nut flour. The bulk density of the flour was similar to that reported by Nina et al. [29] (0.76 g/mL), but higher than that reported by Oladele and Aina [25] (0.55 g/mL) and Yapi et al. [26] (0.5 to 0.54 g/mL). The bulk density value gives an overview on the breakdown of complex molecules like starch [37]. However, a good value might be advantageous in the formulation of functional foods, allowing convenient a package for industrial use [29].

Our tiger nut flour had a dispersibility of around 68% (Table 2), which was nearly comparable to the value of 72.5% reported by Adejuyitan et al. [38] for the same flour type. Furthermore, Ohizua et al. [39] reported a value of 64% for cajan pea flour (*Cajanus cajan*). The dispersibility of a flour is the measure of how its individual molecules disperse and homogenize in a medium. Gaiani et al. [40] define it as a flour-moisturizing quality.

As demonstrated in this work, the absorption capacity of water in tiger nut flour (Table 2) was low, less than that (1.74 g/mL) reported by Nina et al. [29]. This characteristic is believed to be an important technological parameter for controlling the consistency of the dough. It reflects its hydration capacity in the presence of liquid water, and depends mainly on moisture and the rate of damage of the starch. The water absorption capacity also depends on the molecular structure and chemical composition of the starch. The low solubility in starch is likely related to the low proportion of A chains and the high proportion of B1 and B2 chains of amylopectin [41].

The oil absorption capacity (AHC) of the studied tiger nut flour (Table 2) was 2.2 mL/g, higher than the values reported by Nina et al. [29] (1.75 ± 0.02 mL/g), Oladele and Aina [25] (1.13 mL/ g), and Yapi et al. [26] (1.67 to 1.88 mL/ g). The AHC of flour is an important quality in food preservation, as it prevents the development of rancidity [42].

The foaming capacity of tiger nut flour (Table 2) was also high, more than the value in Nina et al. [29] (4.72 ± 1.34%). On the other hand, it was close to that calculated by Oladele and Aina [25] (11.07%). High foaming capacity could result in high levels of starch and protein [37]. Foaming aptitude is desirable in food products such as cakes, bread, meringues, crackers, ice cream, and several other bakery products to maintain a stable texture and structure throughout processing and storage [43].

According to Culetu et al. [44], white and brown rice, maize, oatmeal, millet, amaranth, quinoa, chick-peas and tiger nuts represent gluten-free flours. They are commonly used in gluten-free cereal dietary products because of their nutritional properties. Wet gluten ranges from 17.35 to 32.20% in hard wheat flour [45].

Starches can be classified as A-type (cereals), B-type (tubers), or C-type (leguminous and certain seeds) based on XRD patterns. The C-type pattern combines A and B polymorphs in various proportions and depends on the type of legume [46].

In tiger nut flour, amorphous and crystalline peaks were noted. The three crystalline peaks that occurred in the range 10° to 25° 2θ represented well the semi-crystalline characteristic of B-type tiger nut starch [47].

B chains are resistant starches (RSs), which are the fractions of starch that are not digested in the small intestine but ferment in the large intestine. The end-products of RS fermentation can help reduce cholesterol levels and the risk of colon cancer, in addition to having a prebiotic effect by enhancing probiotic viability [48].

Regarding the supposed pharmacological effect of tiger nuts, two types of extraction were used: 100% aqueous and 30v/70v ratio of water/ethanol, based on previous works and literature. Our results confirmed that a quantity of water is needed for the extraction of phenolic compounds from flour. Our results demonstrated that hydrophilic elements in higher quantities pass into the water as a pure solvent compared to a mixture of water/ethanol 30v/70v. In general, the data expressed in this work were consistent with those of Djikeng et al. [49], who calculated a total polyphenol content ranging from 18.31 to 300.44 mg EGA/100 g in methanolic extracts. Similar values (95.2–388.5 mg/100 g) have also been reported by Oladele et al. [50]. Furthermore, Achoribo and Ong [8] found lower values ranging from 3 to 12 mg EGA/100 g using an ethanolic extract. The difference identified in the literature could be ascribed to environmental conditions (climate, storage location, harvest period, temperature, etc.), plant variety, extraction process, and type of solvent of extraction.

Remarkably, flavonoids were almost equal in both types of aqueous and ethanolic extracts. Xie et al. [51] observed that phytic acid (myo-inositol hexaphosphate, IP6) increases the solubility of flavonoid components after soaking in water and their permeability through the intestinal epithelium.

The pharmacological activities of plants greatly depend on the extraction method being employed [52]. Solvent extraction has widely been used to recover and isolate bioactive molecules as well as in the evaluation of their in vitro activities [53].

For both extracts, antioxidant and antimicrobial activities were linked to the phenolic composition and were also satisfactory. The hydroxyl groups on the flavonoids are mainly responsible for their basic antioxidant activities.

Inhibition zones from Appendix A were consistent with the work of Nwosu et al. [1]. The authors recorded 10 mm for *Shigella* sp., 14 mm for *Salmonella* sp. and *Staphylococcus aureus*, and 16 mm for *Escherichia coli* using cold ethanolic extract of tiger nuts at the same tested concentration (100 mg/mL).

*Cyperus esculentus* L. contains a variety of active ingredients and most of its extracts have an antibacterial effect. Prakash and Ravagan [54] prepared various tiger nut extracts using different solvents such as acetone, 50% ethanol, and chlorophorm. Their antibacterial activities against several pathogens (*E. coli*, *S. aureus*, *Salmonella* sp, *Klebsiella pneumoniae*, *Proteus vulgaris*, *Pseudomonas aeruginosa*, and *Citrobacter freundii)* have been demonstrated using the disc diffusion method. The acetone extract showed the highest inhibitory activity against *S. aureus*, *K. pneumoniae*, and *P. vulgaris.*

The results from the second part of this work describing the technological criteria have permitted us to bake two types of cookies: sugar-free, 100% tiger nut flour cookies, along with cookies soaked in 5% honeybee-based syrup as a carrier of the probiotic strain *L. rhamnosus* SL42. Our results also showed that the physical characteristics of the cookies did not reflect much difference in weight, diameter, thickness, and spread ratio between the plain and the honey–probiotic cookie. In addition, both were manufactured in accordance with international standards compared to commercial cookies.

The spreading ratio and diameter are two criteria used to determine the quality of the flour used and the absorption capacity of the dough. The higher the spreading ratio of the biscuit, the more desirable it is [55]. In the case of cookies, it should be close to 10. 

The cracking resistance of cookies has been linked to the crystalline nature of starch and the denatured proteins [56]. Oddly, and in this study, both baked cookies showed a good resistance to cracking, thus expressing less concern for manufacturers when defining the appropriate packaging that offers greater protection. 

In a recent study, Babiker et al. [57] demonstrated that tiger nut biscuit formulation is acceptable with or without wheat flour.

The results obtained at the first tasting showed that the criteria of sensory quality, taste, color, and smell, as well as the general acceptability of cookies enriched with honey syrup are higher than those of control cookies. After 15 days of storage, the appreciation was reversed in favor of control cookies.

The trichromatic analysis by CIE LAB of plain tiger nut cookies or tiger nut cookies with honey deciphered the dark color of the flour. Color is the most important part of food presentation, since it influences consumer approval of the product. Many reactions can affect color during the processing of foods and their derivatives. The most common are color degradation, browning (such as the Maillard reaction), and ascorbic acid oxidation. Because L*, a*, and b* coordinates are easier and faster to assess than chemical analysis, they can be used indirectly to determine food color changes. Simanca-Sotelo et al. [10] found similar results when studying dietary cookies made with yacon flour (L* = 61.7). Chauhan et al. [56] calculated L* = 65.2 for wheat flour cookies. The protein concentration of a cookie is negatively correlated with its lightness, indicating that the Maillard reaction plays an important part in color development. Chauhan et al. [56] measured chromatic characteristics a* and b* for wheat flour cookies, of 6.3 and 21.8, respectively.

The CIE 1976 trichromatic measurement (L*, a*, b*) is commonly used to analyze the effect of maturity and drying mode on flour color. The L* values represent the flour’s shine. The greater the value of L*, the clearer the flour. Although brown flour made from tiger nut tubers is common and expected, Algerian customers do not generally choose dark brown meal. The acceptance of a cuisine is heavily culturally influenced. This hue could also be generated by browning while baking cookies.

The color development could also be linked to the Maillard reaction, which occurs when the product’s carbohydrates and proteins react, resulting in a brown color. This development is also affected by the cooking time and temperature, as well as the humidity level in the oven.

The cookie is generally regarded as microbiologically stable due to its low initial moisture content and water activity. Changes that impact its quality after storage are primarily linked to texture, specifically the loss of hardness and crustiness, and in certain cases, lipid oxidation [58]. The results of the hedonic test showed that all of our panelists judged our cookies stored for 15 days as having the taste of the product, and no negative connotations were recorded.

Regarding the color characteristic, cookies soaked with honey syrup were darker than the plain cookies, even though this difference was expected during storage.

The texture analysis revealed differences in hardness values between the two sample categories, but this decreased during storage, resulting in similar evaluations among panelists after the 15th day of storage. The texture changes in cookies have been linked to changes in moisture content and interactions between the various components found in the cookie matrix. These texture alterations have a significant impact on the final product’s quality and acceptance [58].

The hedonic test enabled us to distinguish between plain cookies and those fortified with honey and probiotics. It is a good test of appreciation for founding an opinion on the general acceptability of a food product. In general, sensory analysis of our stored cookies revealed that most of our diabetic tasters “like” the cookie made from tiger nut flour. Similarly, Dada et al. [59] showed that the higher the quantity of tiger nut in the composite flour, the better the appearance, aroma, taste, crispness, and overall acceptability of the baked biscuit. Oladunjoye and Alade [60] found that composite cookies were preferred over control sorghum cookies, particularly with a 20% addition of tiger nut pomace, and that the sensory profile obtained was above average.

Regarding the health impact of the baked cookies, tiger nut functional cookies have the potential to provide adequate nutrient requirements, and also to promote good health status. In an interesting study by Oluwajuyitan and Ijarotimi [61], an experimental dough meal containing exclusively tiger nuts reduced diabetic-induced rats’ blood glucose levels by 60.5%. Several tiger nut minerals have also been shown to have health-promoting properties against serious disorders such as high blood pressure.

*L. rhamnosus* growth capacities in the presence of tiger nut flour or honeybee syrup were more than satisfactory. Those tests were needed in order to rule out the possibility of any adverse effects on the probiotic bacteria. The bacterial survival in the stored cookies exhibited a value of 6.43 Log CFU/one cookie when stored for 21 days at 25 °C. This load was appreciated to be highly sufficient in a dry environment, such as that of cookies. Our findings demonstrate the protected probiotic isolate’s strong ability to survive in a moderately dry environment. Its load did not drop considerably (*p* > 0.05) from the amount initially incorporated into each cookie and remained at the recommended level [62]. The trend of adding probiotics to food has already reached the bakery business, with certain bakeries now serving bread, biscuits, crepe mix, muffins, and probiotic-enriched bars. To guarantee the long-term viability of bakery products, probiotics are frequently added after baking. Probiotic formulations could be sprayed over the product at the final stage before packing or integrated in icing cream or chocolate, added after baking.

Several studies were conducted in order to deliver CBBF containing probiotic microorganisms. The review of Mani-López [11] summarized effectively the most valuable research about probiotic survival after baking, but not during storage. Unlike some breads and cereal bars, the probiotics in cakes lost more than the half of the initial incorporated load. High survival counts of probiotic were only found in edible films applied on CBBF after baking or partially baking. Furthermore, cookies with probiotics in coatings were developed. However, the bacterial viability was not evaluated after coating formation.

Adding health-promoting compounds, such as probiotics, might boost a product’s value, but this should not come at the expense of its look, taste, aroma, texture, or other organoleptic attributes.

Finally, SEM analysis decrypted different changes between the two types of cookies. Those differences could be attributed to the partial gelatinization of starch granules exposed to high temperatures and moisture, resulting in a rough surface, severe morphological breakdown, and adherence of other substances to the starch [63]. Overall, the method resulted in more severe starch breakage in small particle sizes, and small spherical structures were more vulnerable to damage. The degree of breakage of the starch types may also be related to the concentration of non-starchy components such as proteins [63].

SEM analysis also showed the persistence and the adherence on the cookie’s surface of a microencapsulated probiotic strain (non-sporing bacterium).

Enhancing probiotic survival is crucial for efficient probiotic production. Spray drying is favored for probiotic lactic acid bacteria powders, offering convenience in handling and storage [64]. Microencapsulation of probiotics with suitable materials has been demonstrated to enhance their viability. Compared to biopolymers, sucrose stands out as a valuable encapsulant, and protects cells against membrane phase transitions and water lost during dehydration. In a recent study, Sompach et al. [24] demonstrated that using sucrose alone as an encapsulant resulted in a reduced fluctuation in membrane fluidity, a key factor protecting membrane integrity and resulting in high cell survival with little cell injury. Similarly, sucrose was able to connect with the polar head group of cell membrane phospholipids via hydrogen bonds, replacing the loss of a water molecule and avoiding membrane phase shift during dehydration [65].

As a result, the cellular damage caused by both dehydration and heat stress could be decreased. Likewise, Sompach et al. [24] reported that sucrose increased the survival rate of *L. reuteri* KUB-AC5 compared to whey protein isolate.

Milk and its derivatives have long been used as probiotic carriers. However, subsequent studies have advocated for the use of plant or food matrices [66]. In a recent study [67], the authors demonstrated the ability of infused *Lactobacillus* cells to survive in a 3D apple scaffold (coated in alginate biopolymer) that had been dried for 2 h in a vacuum-drying oven at 30 °C. The survival was attributed to the probiotic bacteria’s enhanced attachment to the scaffold tissue. This work demonstrated the feasibility of a food-grade delivery system for probiotic cells based on a plant matrix.

## 5. Conclusions

Given the rising consumer demand for healthier snack options, cookies developed with improved nutritional profiles can attract health-conscious customers. Future research directions are needed to improve the sensory attributes, for example, through natural flavors, flour types, and baking techniques. The use of natural preservatives and advanced packaging could also be tested to extend the shelf-life of the products. Finally, conducting sensory evaluations and market studies to align products with consumer preferences would be useful to the define commercial potential of the cookies. By addressing these areas, future research can significantly contribute to the development of commercially viable, nutritionally enhanced cookies that meet consumer expectations while also ensuring a longer shelf-life and appealing sensory attributes.

Further scientific research is needed to determine the stability of encapsulated probiotics in cereal merchandise. To better understand how probiotics interact with the matrix, withstand processing, and retain viability, future studies should take into account various plant-derived matrices, processing settings, other bacterial protective techniques for extended shelf-life, and long storage times. Meanwhile, our findings are considered preliminary and may be used to design future in vitro and in vivo investigations targeted at highlighting the positive effects of both aqueous and ethanolic extracts of tiger nut tubers. This study could also be regarded as the first attempt to make probiotic-containing dietary cookies for diabetics. In the near future, it would be necessary to supplement it with other technological, biochemical, and in vivo nutritional features of these cookies made with tiger nut flour.

## Figures and Tables

**Figure 1 foods-13-02541-f001:**
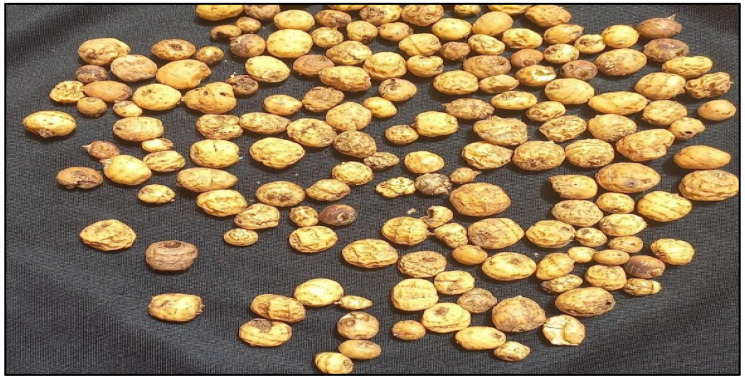
Organic *Cyperus esculentus* L. tubers used in this study.

**Figure 2 foods-13-02541-f002:**
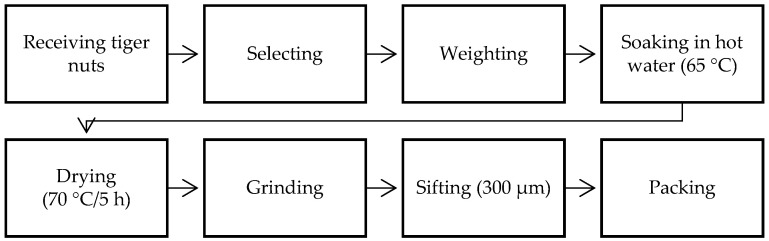
Diagram of production of tiger nut flour.

**Figure 3 foods-13-02541-f003:**
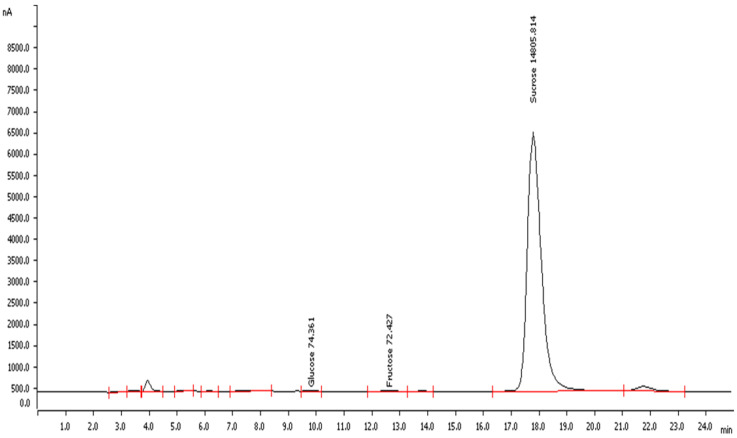
Ion chromatogram of tiger nut’s principal sugars. Peak 1: glucose 74.36 mg/100 g, peak 2: fructose 72.43 mg/100 g, and peak 3: sucrose 14.80 g/100 g.

**Figure 4 foods-13-02541-f004:**
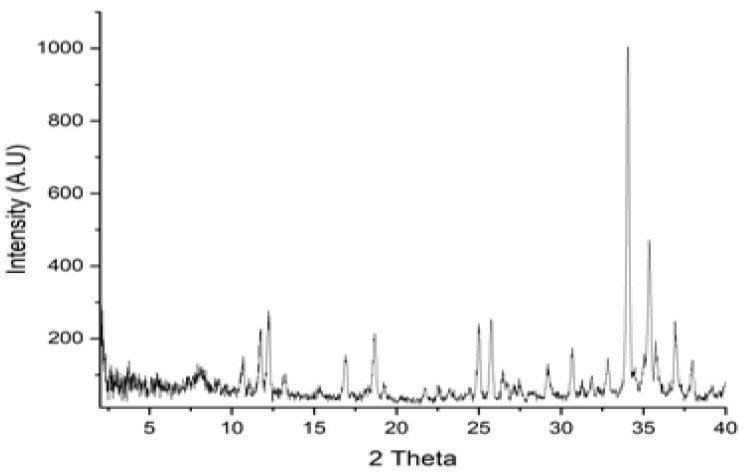
XRD diffraction of the whole organic tiger nut flour.

**Figure 5 foods-13-02541-f005:**
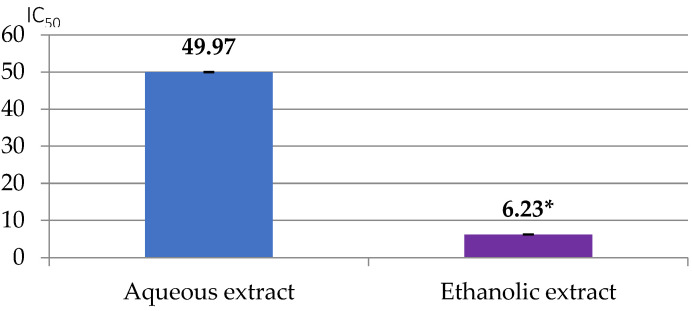
IC_50_ values of tiger nut extracts as calculated with the DPPH method. Aqueous extract is 100% water, and ethanolic extract is 30v water:70v ethanol. * (*p* < 0.05).

**Figure 6 foods-13-02541-f006:**
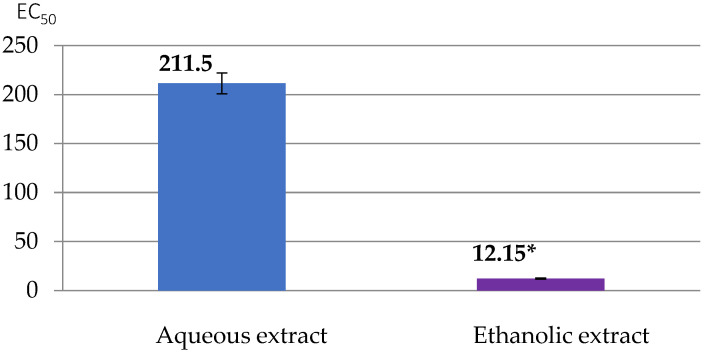
EC_50_ values of tiger nut extracts as calculated with the FRAP method. Aqueous extract is 100% water, and ethanolic extract is 30v water:70v ethanol. * (*p* < 0.05).

**Figure 7 foods-13-02541-f007:**
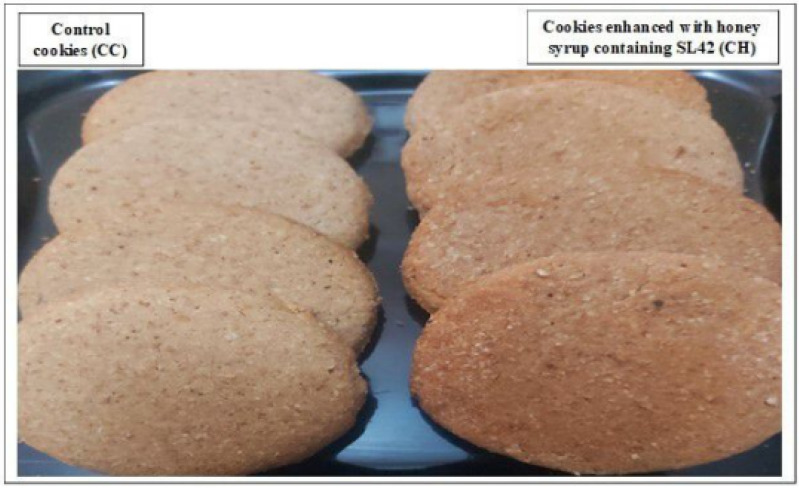
General appearance of the baked cookies. Sugar-free cookies made of 100% tiger nut flour: plain (CC) or enriched with honey syrup containing probiotics (CH).

**Figure 8 foods-13-02541-f008:**
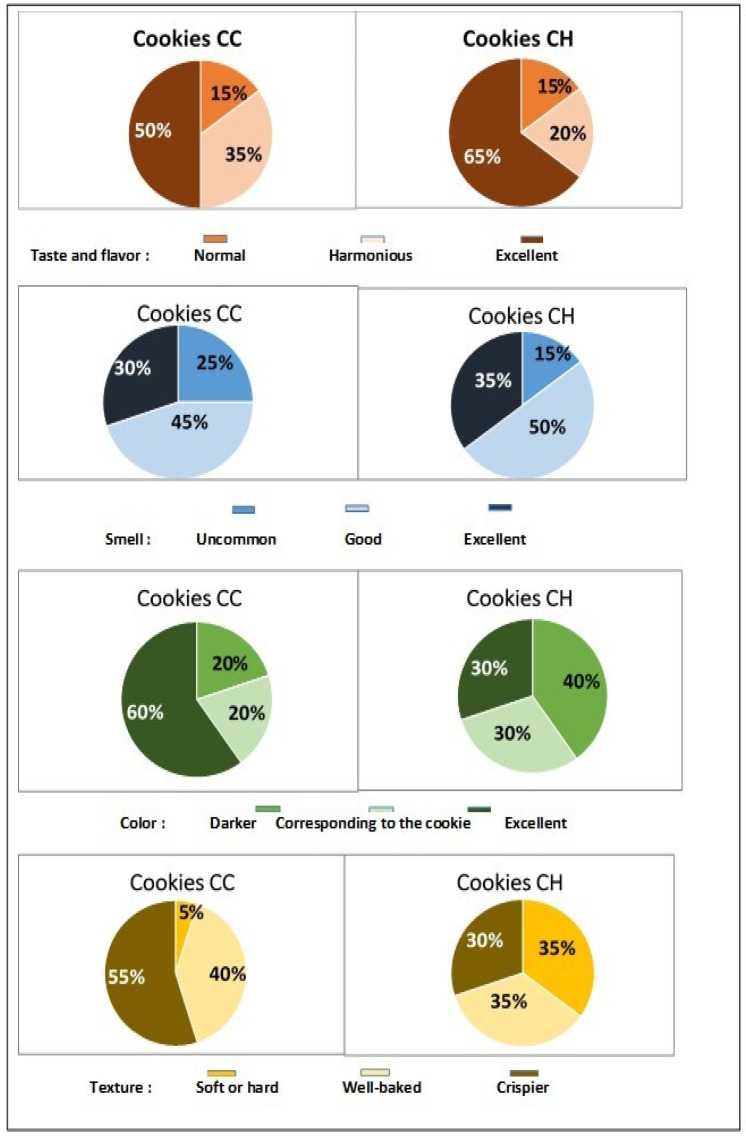
Panelists’ satisfaction level (%) for the taste and flavor, smell, color, and texture of stored plain (CC) or honey-enriched (CH) tiger nut cookies (1^st^ tasting, D = 1 day).

**Figure 9 foods-13-02541-f009:**
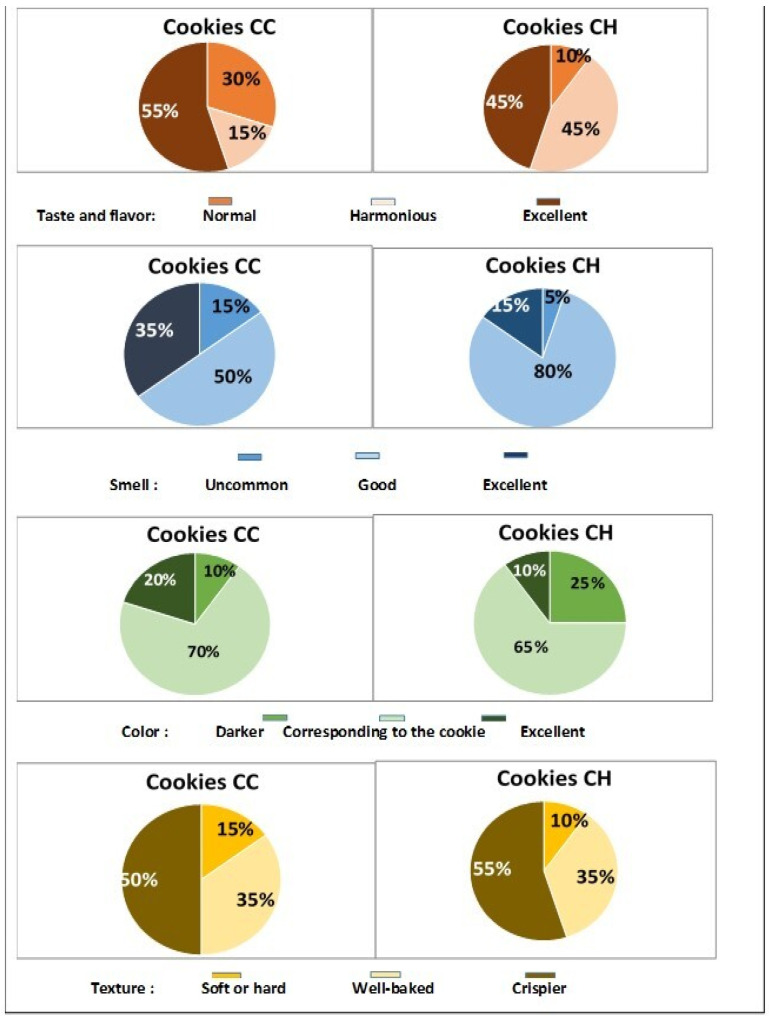
Panelists’ satisfaction level (%) for the taste and flavor, smell, color, and texture of stored plain (CC) or honey-enriched (CH) tiger nut cookies (2^nd^ tasting, D = 15 days).

**Figure 10 foods-13-02541-f010:**
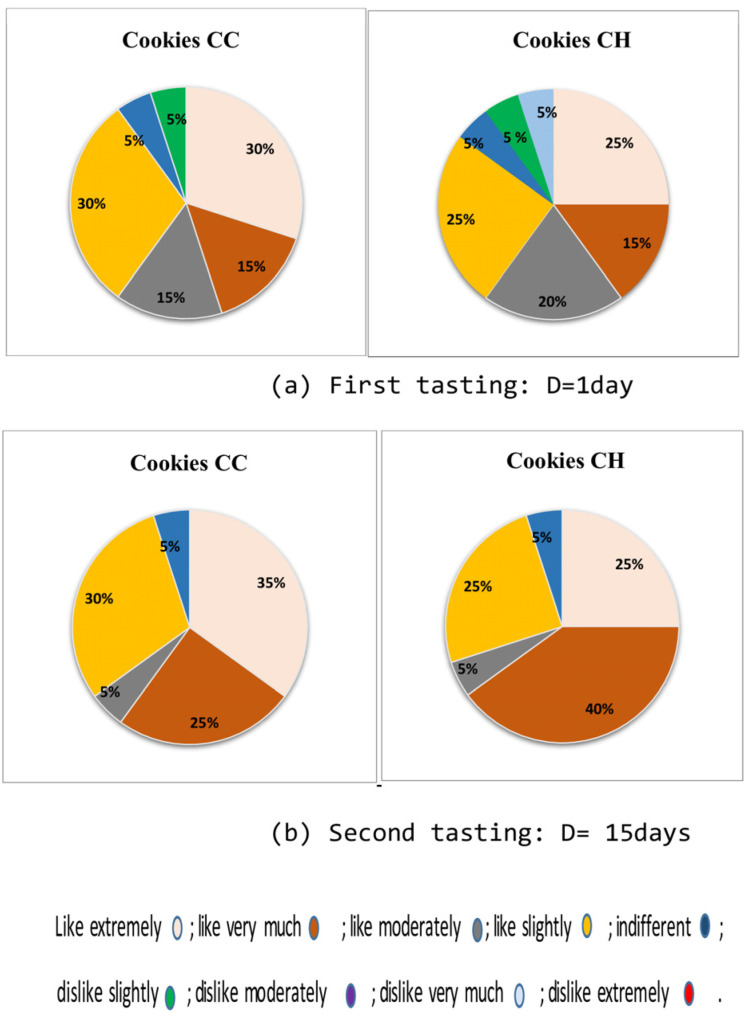
Hedonic test (%) performed on stored plain (CC) or honey-enriched (CH) tiger nut cookies ((**a**): 1^st^ tasting, D = 1 day; (**b**): 2^nd^ tasting, D = 15 days). Nine-point scale was used.

**Figure 11 foods-13-02541-f011:**
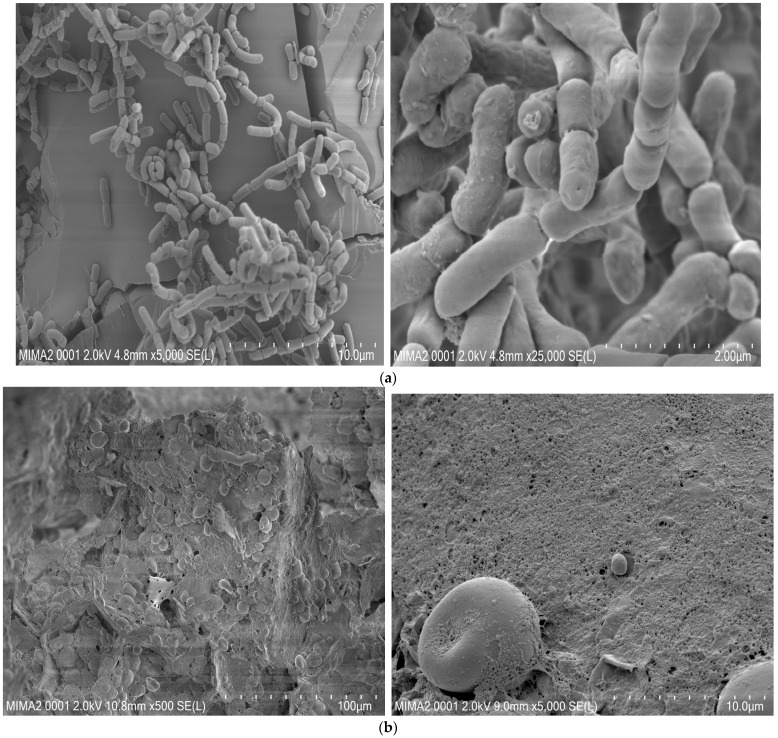
SEM micrographs of (**a**) fixed honey bee syrup containing probiotic strain SL42 at 5000× and 25,000×; (**b**) fixed plain control cookies (CC) at 500× and 5000×; (**c**) fixed cookies enriched with honey bee syrup containing *L. rhamnosus* SL42 (CH) at 500×, 1000×, and 10,000× after 7 days’ storage. *White arrow indicates strain SL42*.

**Table 1 foods-13-02541-t001:** Chemical composition (%) of tiger nut flour.

	Water Content	Ash	Lipids	Carbohydrates	Fibers	Proteins	Minerals
Tiger nut flour	8.3 ± 0.02	3.04 ± 1.2	26.5 ± 1.2	55.6 ± 0.5	15.5 ± 0.01 ^1^	5.8 ± 1.5	2.26 ± 0.05

Means (±standard deviation), *n* = 3, ^1^ g/100 g.

**Table 2 foods-13-02541-t002:** Technological attributes of tiger nut flour.

	Bulk Density (g /mL)	Dispersibility (%)	Water Absorption Index (g/mL)	Oil Absorption Capacity (mL/g)	Foaming Capacity (%)	Wet Gluten Yield (%)
Tiger nut flour	0.74 ± 0.01	68 ± 0.2	0.086 ± 0.2	2.2 ± 0.03	12 ± 0.05	0 ± 0

Means (±standard deviation), *n* = 3.

**Table 3 foods-13-02541-t003:** Physical standards ^1^ of tiger nut cookies compared to the commercial one.

Parameter	Control Tiger Nut Cookie (CC)	Tiger Nut Cookie with Honey Syrup (CH)	Commercial Cookie Made of Wheat Flour
Weight (g)	12.33 ± 0.81 ^b^	12.66 ± 0.51 ^b^	11.7 ± 0.4 ^a^
Diameter (mm)	50.66 ± 0.82 ^a^	50.5 ± 0.54 ^a^	53.85 ± 0.1 ^a^
Thickness (mm)	5.16 ± 0.4 ^a^	5.33 ± 0.51 ^ab^	5.71 ± 0.02 ^b^
Spread ratio	9.81 ± 2.05 ^b^	9.47 ± 1.05 ^a^	9.43 ± 0.1 ^a^
Hardness (N)	97.5 ± 0.3 ^a^	98.3 ± 0.4 ^b^	99.5 ± 0.5 ^c^

^1^ Mean values ± SD of 6–10 cookies. ^a–c^: Significant differences in the same row.

**Table 4 foods-13-02541-t004:** The color characteristics of sugar-free cookies ^1^ made of 100% tiger nut flour: plain (CC) or enriched with honey syrup containing probiotics (CH), stored at 25 °C (1st tasting day: D = 1 day).

Parameter	Control Tiger Nut Cookie (CC)	Tiger Nut Cookie with Honey Syrup (CH)
L*	62.25 ± 4.66 ^a^	61.75 ± 4.94 ^a^
a*	6.95 ± 0.88 ^a^	7.1 ± 0.85 ^b^
b*	26.5 ± 4.61 ^b^	23.5 ± 4.62 ^a^

^1^ Mean values ± SD of 30 cookies. ^a,b^: Significant differences in the same row. L*: lightness; a*: redness; b*: yellowness.

**Table 5 foods-13-02541-t005:** The color characteristics of sugar-free cookies ^1^ made of 100% tiger nut flour: plain (CC) or enriched with honey syrup containing probiotics (CH), stored at 25 °C (2nd tasting day: D = 15 days).

Parameter	Control Tiger Nut Cookie (CC)	Tiger Nut Cookie with Honey-Syrup (CH)
L*	62.75 ± 4.66 ^a^	61.75 ± 4.43 ^a^
a*	7.11 ± 0.48 ^a^	7.17 ± 0.96 ^a^
b*	25.25 ± 4.72 ^b^	23.25 ± 3.35 ^a^

^1^ Mean values ± SD of 30 cookies. ^a,b^: Significant differences in the same row. L*: lightness; a*: redness; b*: yellowness.

**Table 6 foods-13-02541-t006:** Sensory properties of sugar-free cookies ^1^ made of 100% tiger nut flour: plain (CC) or enriched with honey syrup containing probiotics (CH), stored at 25 °C (1st tasting day: D = 1 day).

Quality Parameter	Control Tiger Nut Cookie (CC)	Tiger Nut Cookie with Honey Syrup (CH)
Taste	5.5 ± 2.01 ^a^	6.1 ± 1.86 ^b^
Color	3.35 ± 1.53 ^a^	3.75 ± 1.58 ^b^
Aroma	3.7 ± 1.17 ^a^	3.85 ± 1.03 ^a^
Texture	1.45 ± 0.60 ^a^	1.45 ± 0.82 ^a^
Overall acceptability	3.5 ± 1.32 ^a^	3.66 ± 1.32 ^a^

^1^ Mean values ± SD of 100 cookies. ^a,b^: Significant differences in the same row.

**Table 7 foods-13-02541-t007:** Sensory properties of sugar-free cookies ^1^ made of 100% tiger nut flour: plain (CC) or enriched with honey syrup containing probiotics (CH), stored at 25 °C (2nd tasting day: D = 15 days).

Quality Parameter	Control Tiger Nut Cookie (CC)	Tiger Nut Cookie with Honey Syrup (CH)
Taste	5.4 ± 2.25 ^a^	5.55 ± 1.63 ^a^
Color	3.75 ± 0.91 ^b^	3.15 ± 1.26 ^a^
Aroma	3.95 ± 1.19 ^b^	3.3 ± 1.12 ^a^
Texture	1.35 ± 0.74 ^a^	1.45 ± 0.68 ^a^
Overall acceptability	3.61 ± 1.27 ^b^	3.36 ± 1.17 ^a^

^1^ Mean values ± SD of 100 cookies. ^a,b^: Significant differences in the same row.

**Table 8 foods-13-02541-t008:** Survival of the *L. rhamnosus* SL42 probiotic strain (Log CFU/one cookie ^1^) in cookies made from tiger nut flour and enriched with 5% honeybee syrup (CH) during three weeks’ storage at 25 °C.

Storage Period	Tiger Nut Cookie with Honey Syrup (CH)
Day 0	7.15 ± 0.43 ^a^
Day 1	6.93 ± 0.96 ^a^
Day 7	6.66 ± 0.44 ^a^
Day 15	6.55 ± 0.35 ^a^
Day 21	6.43 ± 0.05 ^a^

^1^ one cookie = 12 g. ^a^: non-significant differences in the same column.

## Data Availability

The original contributions presented in the study are included in the article/Appendix A, further inquiries can be directed to the corresponding author.

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
