# Peer review of "Functional Cyperus esculentus L. Cookies Enriched with the Probiotic Strain Lacticaseibacillus rhamnosus SL42"

_foods, 2024, doi:10.3390/foods13162541_

Round 1

Reviewer 1 Report

Comments and Suggestions for Authors

General Comments

The manuscript is well-organized and provides a comprehensive analysis of the tiger nut cookies, addressing both the physico-chemical properties and sensory qualities. The use of a probiotic strain and the focus on sugar-free formulations align with current trends in health-conscious food products. However, there are several areas that require clarification and improvement.

Specific Comments

Introduction: The introduction provides a good background on tiger nuts and the importance of probiotics. However, it could benefit from a clearer statement of the study's objectives and the specific research questions being addressed.

·         Clearly outline the study objectives and hypotheses. Provide more context on why tiger nut flour was chosen over other gluten-free flours.

Materials and Methods: The methods section is detailed but lacks clarity in some areas, particularly concerning the preparation of the cookies and the specific analytical techniques used.

·         Provide a detailed description of the cookie preparation process, including the proportion of ingredients used.

·         Specify the analytical methods and instruments used for measuring the physico-chemical properties, and cite relevant protocols or standards.

·         The specific sensory attributes evaluated (e.g., taste, texture, aroma) and the hedonic scale used.

·         The exact composition and preparation process of the honey syrup used for probiotic enrichment.

Results and Discussion: The results are well presented with clear tables and figures. However, the discussion could be more critical, particularly in comparing the results with existing literature.

·         Include a more in-depth comparison of the physico-chemical and sensory properties of the tiger nut cookies with those reported in the literature for similar gluten-free or probiotic-enriched products.

·         Discuss the implications of the findings in the context of consumer health, especially focusing on the potential benefits of the high mineral content and probiotic viability.

Conclusion: The conclusion summarizes the key findings but does not clearly state the potential applications or future directions of the research.

·         Highlight the practical applications of the research, such as potential commercial production of the cookies, and suggest future research directions, particularly in improving the sensory attributes and shelf-life of the products.

Figures and Tables: The figures and tables are generally well-prepared, but some lack sufficient explanatory captions.

·         Ensure that all figures and tables have comprehensive captions that explain what is being shown and provide context for interpreting the data.

·         Improve the quality of images, particularly Figures 8-10. Ensure that they are high-resolution and clear to facilitate accurate interpretation. High-quality images are crucial for presenting detailed data and visual information effectively.

Abbreviations: Define all abbreviations upon first use in the manuscript.

Comments on the Quality of English Language

While the overall language quality is understandable, some sections are overly verbose and could be made more concise. Additionally, there may be minor grammatical errors that need addressing.

Author Response

Dear reviewer 1,

our responses are detailled in the file.

Regards,

Reviewer 2 Report

Comments and Suggestions for Authors

The article is interesting. However, it needs a few corrections:

1. The introduction lacks a description of why the use of probiotic strain Lacticaseibacillus rhamnosus SL42 in cookies is new. Have similar cookies been made before?

2.Table 1 needs to be corrected, something has moved in the file. Please add statistics to table 1 and 2.

3. Figure 5 and 6, please complete the title of the axis.

4. Table 3-8, letters should be in superscript.

5. Funding - "The APC was funded by XXX." - description missing, or please delete.

6. Please correct your literature citation according to the instructions for authors.

Author Response

Dear reviewer 2,

our responses are detailled in the file.

Regards,

Reviewer 3 Report

Comments and Suggestions for Authors

The study entitled Functional Cyperus esculentus L. Cookies enriched with the Probiotic strain Lacticaseibacillus rhamnosus SL42 dealt with testing dietary cookies made from organic tiger nut flour with the addition of honey-bee syrup containing a probiotic strain SL42.

The need for new, innovative solutions in the production of functional food has experienced an expansion in recent years, and these and similar studies are essential. Nevertheless, it is necessary to state the global importance of the selection of this raw material, various supplements, and the importance of the selected probiotic strain.

The abstract is very confusingly written and I'm not sure what the point of this study is.

The introduction has to be rewritten because 12 references failed to show the current state-of-the-art and the basic data necessary to introduce the scientific community to the objective of the investigation. Additionally, the authors do not mention the importance of lactic acid bacteria, probiotics, LAB, and lactobacilli in any sentence. What is the purpose of the lactobacilli supplement, why SL42?

The authors used different methods, but they must be written so that other researchers can reproduce these methods.

The authors began the Conclusion with the sentence "Further scientific research is needed to determine the stability of encapsulated probiotics in cereal merchandise." instead of pointing out what was obtained in this study.

In Figures 5 and 6, I am unsure what is being observed (the y-axis is not marked), except that there is almost no standard deviation.

Figures 8, 9, and 10 represent the Panelists' satisfaction and it isn't easy to follow what the Panelists' conclusion is.

I suggest rewriting this study.

Author Response

Dear reviewer 3,

our responses are detailled in the file.

Regards,

Round 2

Reviewer 2 Report

Comments and Suggestions for Authors

The authors revised the manuscript according to suggestions.

Reviewer 3 Report

Comments and Suggestions for Authors

After the first round of review, the quality of the manuscript is improved slightly.